# The Role of Endothelial Cells in the Onset, Development and Modulation of Vein Graft Disease

**DOI:** 10.3390/cells11193066

**Published:** 2022-09-29

**Authors:** Shameem S. Ladak, Liam W. McQueen, Georgia R. Layton, Hardeep Aujla, Adewale Adebayo, Mustafa Zakkar

**Affiliations:** Department of Cardiovascular Sciences, Clinical Science Wing, University of Leicester, Glenfield Hospital, Leicester LE3 9QP, UK

**Keywords:** endothelial dysfunction, vein graft disease, intimal hyperplasia, accelerated atherosclerosis, surgical and clinical management

## Abstract

Endothelial cells comprise the intimal layer of the vasculature, playing a crucial role in facilitating and regulating aspects such nutrient transport, vascular homeostasis, and inflammatory response. Given the importance of these cells in maintaining a healthy haemodynamic environment, dysfunction of the endothelium is central to a host of vascular diseases and is a key predictor of cardiovascular risk. Of note, endothelial dysfunction is believed to be a key driver for vein graft disease—a pathology in which vein grafts utilised in coronary artery bypass graft surgery develop intimal hyperplasia and accelerated atherosclerosis, resulting in poor long-term patency rates. Activation and denudation of the endothelium following surgical trauma and implantation of the graft encourage a host of immune, inflammatory, and cellular differentiation responses that risk driving the graft to failure. This review aims to provide an overview of the current working knowledge regarding the role of endothelial cells in the onset, development, and modulation of vein graft disease, as well as addressing current surgical and medical management approaches which aim to beneficially modulate endothelial function and improve patient outcomes.

## 1. Introduction

Ischaemic heart disease (IHD) is the single biggest killer worldwide. In the UK, there are more than 100,000 hospital admissions each year due to heart attacks. Furthermore, heart and circulatory diseases cause more than a quarter of all deaths in the UK with an average of 450 deaths each day [1]. Management of IHD generally involves the modification of risk factors such as smoking, hypertension, and diabetes mellitus. However, more invasive treatment is required in a subgroup of patients who will need revascularisation to restore blood flow in the coronaries where there is significant stenosis. Two main revascularisation approaches are available—percutaneous intervention (PCI) and coronary artery bypass graft (CABG). The first procedure relieves the occlusion of the coronary artery and improves blood supply to the ischaemic tissue by using a combination of balloon angioplasty to compress the plaque area, prior to stenting of the affected area to minimise re-occlusion [2,3]. Alternatively, CABG restores blood flow by using conduits to bypass the diseased area [4]. Studies have identified that CABG is associated with better health outcomes versus PCI, particularly in cases of more significant disease burden and patients with specific comorbidities such as diabetes mellitus [5].

The main conduits of choice for CABG are the left internal mammary artery (LIMA), the radial arteries (RA) and the long saphenous vein (LSV). The use of arterial grafts is preferred owing to their greater long term patency rates; however, its use is limited due to technical aspects such as vessel length and degree of stenosis in native vessels [4]. Thus the LSV is the commonly used due to its ease of accessibility and significant length, allowing for the construction of several grafts from the same vessel [6]. However, there remains a significant drawback to the long-term patency of this graft with a significant rate of failure (50–60%) within 10 years post-operation [7,8]. The pathology driving this high rate of failure is termed ‘vein graft disease’, which derives from the onset and development of intimal hyperplasia (IH) and accelerated superimposed atherosclerosis within the graft, resulting in occlusion and renewed symptoms of ischaemia [9,10]. One major contributor to the onset and progression of vein graft disease is believed to the endothelial cells (ECs), which comprise the innermost layer of the vasculature. As such, the purpose of this review is to provide insight into the current research investigating the role of ECs in the development of vein graft disease, and current surgical and medical management approaches under investigation for improving the long-term outcomes for patients undergoing revascularisation surgery.

## 2. Differences between Arteries and Veins

In general, veins have a larger diameter than arteries. The outermost layer, tunica externa (or adventitia), is rich in collagen and elastic fibres providing a supporting framework to the vessel. The next layer towards the lumen is the tunica media, composed of smooth muscle cells (SMCs) and elastic fibres that provide muscular support to the vessel and elasticity, respectively. In arteries, this is the thickest layer; however, in veins, the medial layer is thin and not well-defined, making them less rigid or prone to spasm and able to hold a larger amount of circulating blood as compared to arteries. Another striking difference between arteries and veins is the presence of venous valves extending into the lumen to avert the backflow of blood and the absence of elastic lamina that separates tunica media and tunica intima in veins. The latter is the innermost layer of the vasculature featuring the endothelium that directly lines the vessel lumen [11].

There are distinct differences between the arterial and venous endothelium. This includes differences in size and shape—for instance, arterial ECs are generally long, narrow and thicker and the intercellular junctions are much tighter than those in veins [12]. Venous ECs are shorter and wider as venous circulation experiences significantly lower blood flow rate compared to the arterial circulation [13]. Veins in situ experience low shear stress, typically 1–6 dynes/cm^2^, while arteries experience shear stress levels of 10–20 dynes/cm^2^, as well as much lower flow and pressure [14]_._ Under normal physiological conditions, ECs and SMCs have a very low proliferation rate. However, high shear stress promotes vessel wall proliferation and thickening in vein bypass grafts implanted into the arterial circulation [15]. Increased rates of blood-flow-induced mechanical stresses are known to alter venous EC phenotype [16]. Furthermore, studies have shown that ECs can change phenotype in response to stimuli, leading to the activation of pathways such as those involved in remodelling and endothelial to mesenchymal transition (EndoMT) [13,17,18].

The significant plasticity of ECs is derived in large part from their developmental origin and the vast differences in stimuli to which they are exposed, as can be seen in the significant heterogeneity of these cells throughout the vasculature and the body as a whole [19]. Whilst this plasticity is critical in maintaining homeostatic balance, in pathological conditions (such as atherosclerosis) or conditions that deviate significantly from a physiological norm (such as vein graft implantation into arterial circulation), this plasticity may inadvertently exacerbate a disease state [20]. Remodelling of the vein graft is not inherently detrimental—in fact, the goal of the vein graft implantation (beyond revascularisation) is to facilitate the ‘arterialisation’ of the graft, such that it acts and responds to haemodynamic cues in a comparable manner to the native arteries. This ideally involves the outward remodelling of the vein, such that the vessel walls become thicker and better able to maintain their integrity under arterial haemodynamic conditions. However, this process is non-specific, often resulting in the inward remodelling (termed intimal hyperplasia) which facilitates the occlusion of the vessel leading to failure [21]. The complexities surrounding this biological re-engineering is a multifaceted morphological and metabolic process, and as such, understanding the mechanism of this process, and how clinicians/surgeons attempt to modulate this process, is central to ensuring the long-term patency of these grafts and outcomes for CABG patients. (Figure 1).

## 3. Endothelial Cells Function in Vascular Homeostasis and Inflammation

ECs play a pivotal role in regulating vascular homeostasis. Quiescent endothelium protects arteries by providing an anticoagulant, anti-inflammatory environment, and regulating lipoprotein permeability. The endothelium regulates vascular tone by modulating SMC proliferation, migration, and contractile function [22,23]. Maintenance of vascular homeostasis is possible due to the endocrine nature of ECs, ensuring that changes in local haemodynamic can be accounted for via the release of factors such as nitric oxide (NO) and prostaglandin (PGI2) for vasodilation, and factors such as endothelin A and angiotensin II (AngII) for vasoconstriction, among others [24,25]. Upregulation of these factors also prevents platelet adherence and aggregation through increases in their cyclic adenosine monophosphate (cAMP) content by degrading factors responsible for activating platelets [26]. Beyond this, blood fluidity is also controlled by expression of tissue factor pathway inhibitors (TFPIs), heparin sulphate proteoglycans and thrombomodulin, which block coagulation, inactivate thrombin, and modulate thrombin specificity to activate protein C, respectively. Protein C, in concert with protein S, acts to inactivate the clotting signalling pathway [27]. Quiescent ECs also minimise interactions with leukocytes, suppressing the transcription of adhesion molecules (VCAM1, ICAM1, E-selectin), as well as sequestering proteins known to facilitate leukocyte activation (e.g., P-selectin) [28]. The bioavailability of these factors, therefore, is critical in maintaining cell quiescence of the SMCs comprising the medial layer, preventing outgrowth of these cells, ensuring vessel integrity, and maintaining an overall anti-inflammatory, anti-proliferative and anti-coagulant environment [29].

ECs are the primary target of inflammation; thus, its response to this stimulus is central in facilitating or inhibiting signal propagation. EC activation describes the alteration in endothelial properties, leading to an increase in endothelial-leukocyte interactions, which are essential for inflammatory responses [30]. A key event in inflammation is the recruitment of leukocytes from the circulation to the vessel wall via their adhesion to activated ECs. This is coordinated by a process involving several families of adhesion receptor families. The activation of vascular endothelium leads to the rapid expression of members of the Selectin family which coordinate attachment and adherence of leukocytes to ECs [31,32]. One of these, P-Selectin, is stored in Weibel-Palade bodies within the EC and is released by exocytosis within minutes of activation [33]. In contrast, E-Selectin expression in activated ECs requires de novo transcription and protein synthesis. Both P-Selectin and E-Selectin interact with Sialyl LewisX carbohydrate groups expressed by leukocytes and lymphocytes, initiating the rolling of leukocytes along the endothelial surface [34]. In addition to E-Selectin and P-Selectin interactions, L-Selectin expressed on the leukocytes can interact with charged fucosylated carbohydrate groups on the endothelium and contributes largely to the capture of leukocytes during the early phase of recruitment [35,36]. Interactions of leukocytes with the endothelium facilitate their activation by surface bound chemo-attractants (e.g., MCP1). Subsequently, interactions between activated integrins (e.g., VLA4) expressed on leukocytes and their ligands (e.g., VCAM1) expressed on activated ECs reduces the velocity of rolling leukocytes and leads to firm adhesion [34]. Transmigration is the final step in the adhesion cascade and is mediated by adhesion molecules such as platelet endothelial cell adhesion molecule-1 (PECAM1), intercellular adhesion molecule-1 (ICAM1), and junctional adhesion molecules (JAMs) [37,38]. Furthermore, in response to inflammatory stimuli, there is upregulation of NO and PGI2 which diffuses to the medial SMCs triggering vasorelaxation through cyclic guanosine monophosphate dependent pathways. The controlled expression of NO is critical, as decreased expression of NO has been implicated in the upregulation of reactive oxygen species (ROS) production, particularly superoxide (O_2_^−^), generated as a result of uncoupled endothelial nitric oxide synthase (eNOS), and the action of NADPH oxide and xanthine oxidase enzymes [39]. Increased ROS presence not only facilitates vasoconstriction but also acts to scavenge available NO (forming peroxynitrite), which, in turn, can lead to the establishment of an atherogenic, pro-inflammatory environment deriving from EC dysfunction, which is central to numerous vascular disease states [40].

## 4. The Role of Endothelial Cells in Vein Graft Disease

IH is a multifactorial inflammatory process that starts during the harvesting and preparation of veins. Vasospasm, surgical trauma, and ischaemia during harvesting can activate vein ECs. Implantation of the venous segment in the arterial environment promotes EC activation and the expression of different molecules such as ICAM1, VCAM1, MCP1, and IL1, thus triggering the influx of immune cells associated with development of IH and vascular remodelling [16,41,42].

IH begins with EC activation followed by the migration and proliferation of SMCs to the intima and changes in extracellular matrix [43]. Some studies suggest that vein graft ECs are lost early post-surgery [44]. However, most of these studies looked at later time points but did not address the immediate changes in ECs. Furthermore, most studies were conducted using animal models of end-to-end vein–artery interposition grafting [45], which differs from the end-to-side technique used for grafting in CABG. Moreover, ex vivo exposure of veins to arterial flow demonstrated the preservation of ECs or only partial denudation, and that residual ECs were highly activated [42,46,47,48].

Signalling intermediaries such as Nuclear Factor-κB (NF-κB) are implicated in vascular inflammation regulation [42]. NF-κB regulates the leukocyte adhesion cascade, leukocyte transmigration and the release of pro-inflammatory mediators. Previous studies have shown that NF-κB inhibition in vivo [49,50] and ex vivo delivery of an NF-κB decoy can significantly prevent IH after vein grafting [51]. However, these studies focused on alteration of SMC proliferation and migration following NF-κB inhibition, rather than the role of NF-κB in vein graft ECs following implantation into its new haemodynamic environment. Our group recently demonstrated mechanistically the exposure of venous ECs to acute high shear stress is associated with NF-κB pathway activation and that inhibition of this pathway significantly improved endothelial barrier function and reduced pro-inflammatory gene expression and leukocytes adhesion to ECs [42]. In particular, exposure to arterial shear stress in human LSVs resulted in the upregulation of pro-inflammatory cytokine CCL2 expression and enhanced nuclear translocation of the NF-κB p65 sub-units, with inhibition of NF-κB corresponding with induction of monocyte adhesion to the vessel endothelium. In vitro study of ECs corroborated p65 translocation in response to shear stress, downregulated IκBα expression and increased phosphorylation at the Ser-276 residue, Furthermore, targeted inhibition of this pathway was sufficient to mitigate endothelial CCl2 expression and monocyte recruitment and disruption of EC cell–cell contact.

It is known that vein graft arterialisation leads to the activation of p38-mitogen-activated protein kinases (MAPK). However, the exact role of MAPK relative to vein graft disease and their mode of regulation remain largely unknown, as data suggesting MAPK role in the development of vein graft disease are limited. A study utilising a canine model of external jugular vein interposition in the carotid artery showed that MAPK pathway activation is bimodal as p38 activation was observed as early as 30 min and at 3 h post-surgery. Moreover, p38 activation was also noted later at 4 days, which was associated with SMC activation [52]. Additionally, in a rabbit model of vein grafting, the use of topical MAPK inhibitors significantly suppressed disease progression [53]. Another vein graft model utilising porcine carotid jugular interposition graft showed that pharmacological vasorelaxation suppresses p38 activity and limits cell cycle progression and proliferation [54]. Our group has demonstrated that acute high shear stress elevates MKP-1 expression in vitro, and a corresponding decrease in VCAM-1 expression and p38 activation, validated using an MKP-1 silencing approach. Furthermore, we were able to identify preferential expression of MKP-1 at high-shear, atheroprotected sites within murine aortas as determined by fluid dynamic models, and that MKP-1 suppresses the activity of both p38, JNK and the expression of VCAM-1 in these atheroprotected regions. A subsequent study by our group was able to determine that arterial shear stress activates venous ECs, but not arterial ECs, as determined by in vitro upregulation of chemokine MCP-1 and IL-8 expression induced p38 MAPK phosphorylation and confirmed through p38 inhibitor studies. Arterial ECs were revealed to be protected from the shear-induced expression of p38-dependent chemokines as a result of the rapid induction of MKP-1 and JNK (1 h vs. 12–24 h in venous ECs). Silencing MKP-1 resulted in enhanced p38 activation and elevation of the aforementioned chemokines. Moreover, we identified dexamethasone as an effective treatment for the suppressing the proinflammatory activation of venous ECs via the induction of MKP-1 by effectively ‘arterialising’ these cells. Combined, these studies identify that MKP1 induction is critical in explaining the atheroprotective and anti-inflammatory behaviour observed in high shear regions of arteries, but not veins. Therefore, activation of MPK-1 in vein grafts may offer therapeutic potential in preventing the activation of pro-inflammatory responses triggered by MAPK signalling [55,56].

Vascular remodelling in vein grafts is thought to be a normal response to adaptation to vein graft arterialisation which includes high shear stress and stretch force [57]. A key feature of vein graft remodelling is EndoMT, wherein vein ECs lose their endothelial cell characteristics and transition to express markers of SMCs [58]. Several studies have shown the upregulation of transforming growth factor beta (TGFβ), which is implicated in EndoMT, at sites of vascular injury [58,59,60]. There is limited data available on EndoMT in vein grafts; however, an in vivo murine cell lineage-tracing model suggested TGF-β driven EndoMT as an important pathway responsible for neointimal formation. This study also assessed EC and SMC marker expression in histological sections from newly acquired early-phase-failed and long-term patient vein grafts. Data revealed a subset of cells in the neointima co-expressing both endothelial and immature SMC markers in the early phase failed vein grafts suggesting a role of EndoMT in vein graft remodelling. Interestingly, the TGFβ signalling pathway has also been identified as a therapeutic target to prevent vein graft stenosis [58]. Furthermore, increased expression of TGFβ is associated with intimal thickening accompanied by increased collagen in the neointima of rat vein grafts demonstrating its role promoting the development of IH [59].

A particular pathway implicated in modulating endothelial (dys)function is the phosphoinositide 3-kinase/protein kinase B (PI3K/Akt) signalling pathway, which has well-established functions associated with regulation of vascular tone and angiogenesis, alongside the recruitment and control of leukocyte adhesion to the endothelium [61]. This behaviour is directly opposed by phosphatase and tensin homolog deleted on chromosome ten (PTEN), with known functions related to cell migration and apoptosis [62]. A study by Kuo et al. [63] was able to identify that PTEN overexpression and subsequent Akt downregulation was present in atherosclerotic vessels, in turn suppressing the angiogenic potential of the vessel ECs, through the attenuation of endothelin-1/ endothelin B receptor expression, and subsequent disruption of endothelin-1/ NO homeostasis. This modulation of angiogenic signalling implicates the PI3K/Akt pathway, and PTEN in particular, in facilitating endothelial dysfunction and thrombosis within vein grafts.

There is evidence that the process of remodelling in arterialised LSV is accompanied by oxidative stress. In vein ECs, mechanical stretching induces ROS production to a greater extent than arterial ECs as the former is more sensitive to ROS effects [64]. Furthermore, data from several studies suggest the significance of oxidative stress modulation in neointima formation in vein grafts [65,66,67]. During normal conditions, ROS production can occur because of aerobic metabolism in the mitochondria. However, the cells’ inbuilt enzymatic and free radical scavenging mechanisms neutralise these radicals, resulting in a redox or homeostatic state [68]. Under stress conditions, overproduction of ROS in response to various stimuli such as hyperglycaemia and hypertension results in reduction in NO bioavailability. This leads to endothelial dysfunction as ECs produce vasoconstrictor agents thus initiating an inflammatory response and triggering the activation of signalling pathways such as MAPK and NF-κB [69,70]. Use of antioxidants in ECs can shorten the kinetics of MAPK and inhibit induction of E-Selectin and VCAM1 thereby counteracting the impact of ROS [68]. It is common practice for grafts to be stored temporarily in solutions (saline or blood-based) that do not protect against ischaemia-reperfusion injury prior to implantation. A recent study suggests that oxidative damage can be prevented by storing vascular grafts in a preservation solution such as DuraGraft^®^ with sufficient antioxidant activity [71]. All of the aforementioned processes have ben summarised in Figure 2.

## 5. Techniques to Preserve Endothelial Cells in Vein Grafts

### 5.1. Harvesting Techniques

The conventional technique (CT) for vein harvest as initially described by Favaloro et al. [72] entails vein exposure along its length through a longitudinal incision following the contours of the leg and is then harvested in a skeletonised fashion, stripped of surrounding peri-vascular tissue. This induces a degree of venospasm that is subsequently counteracted with manual distention of the vein using crystalloid or blood solutions. Moreover, inflating the vein during the harvest to check for side branches at pressures significantly greater than that of the human vascular system disrupts the endothelium [73], promotes thrombosis [74] and damages the tunica within the vein wall [75]. Damage to the endothelium alters the release of endothelial-derived vasoactive substances including NO, prostanoids, and endothelin-1. In response to damage, the EC production of NO and prostacyclin is reduced but production of prothrombotic factors such as thromboxane and endothelin-1 is increased [76]. Damaged endothelium is pro-coagulant and acute thrombosis is a primary cause of early graft failure [77].

The no-touch technique (NTT) was first described by de Souza in 1996 [78]. Souza proposed that the vein could be harvested with surrounding adventitia, and by doing so, the vein is cushioned by a border of soft tissue which contains the important perivascular structures such as vasa vasorum (VV) and nerves. Importantly, the vein is not directly handled during this process. This technique preserves both the endothelium of the vessel and the microvasculature that nourishes the vein wall. The VV has been shown to maintain perfusion of the vein wall through direct contact with the vein lumen after harvest [79] and demonstrate much greater intimal density and penetration compared to arterial grafts, suggesting a greater significance in the maintenance of luminal viability after harvest. Stripping of the VV by CT therefore promotes ischaemia within the LSV wall with a subsequent detrimental impact on patency, [80] which is likely of larger impact than the associated ischaemia of arteries harvested by CT. The NTT therefore avoids physical disruption to these surrounding tissues, limiting ischaemic injury to the endothelium and provides a physical buffer to limit inadvertent kinking of the graft.

Most of the surrounding ‘cushion’ with which the vein is harvested is composed of fat. Lipocytes have been demonstrated to be a vital source of locally derived vasoactive factors essential for the regulation of vascular tone [81], especially NO. There is compelling evidence that NO expression in vein grafts after CABG plays a key role in the reduction of both IH and atherosclerosis in mouse models [82]. Dashwood et al. [83] demonstrated the presence of eNOS and NO within the periadventitial fat surrounding the LSV following harvest using NTT. This suggests that another benefit of the NTT is not only avoidance of direct physical insults, but also by preservation of the vasoactive sources local to the vein. A further benefit of the NTT relates to the minimised distention of the vein during preparation, which is known to impact on EC integrity, and was previously demonstrated by Angelini et al. [75,84] who confirmed previously that elevated distention pressure causes a reduction in the concentration of adenosine triphosphate (ATP) within the vein and that preservation fluids have the potential to improve ATP levels. The superiority of the NTT was more recently confirmed clinically with higher patency rates of veins harvested by this method as compared to radial arteries or veins harvested conventionally. Furthermore, this also appears to translate into better outcomes as demonstrated by Tian et al. [85] in their multicentre randomised control trial of 2533 patients. It is essential to note that the NTT is more challenging in patients who are obese and when the LSV is very superficial due to anatomical variation [78].

Endoscopic harvesting (EH) is a minimally invasive technique that aims to reduce both graft and wound complications, and to optimise patient acceptability following LSV harvest. The process utilises significantly smaller incisions than CT harvest and uses a camera system within either a pressurised or a non-pressurised CO2 tunnel to dissect the vein from surrounding tissues. The primary clinical benefits are a reduction in early and late postoperative pain and wound infection [86]. When it comes to the impact of EH on the integrity of the cellular components of veins, the picture is different. Hashmi et al. [87] used immunohistochemical analysis to determine EC viability and expression between techniques. They demonstrated greater EC detachment (*p* = 0.01) and much lower preservation of ECs through staining of CD31 antibody in veins harvested with CT compared to closed-tunnel EH (mean grading score 2.6 ± 1.43, *p* = 0.70 for CT versus 3.2 ± 1.23, *p* = 0.03 for closed-tunnel EH). Wheeler et al. [88] quantified EC damage through ex vivo analysis of NO-mediated endothelial dependent relaxation (EDR) of LSV grafts in response to bradykinin after forced contraction. Of the 28 venous segments assessed (11 segments from five EH patients and 17 segments from eight CT patients), EH segments showed greater EDR at every concentration of the bradykinin dose–response curve (*p* = 0.029). They found similar structural preservation in both cohorts, suggesting that whilst rates of structural damage may be similar between techniques, preservation of EC function is improved with EH.

In direct contrast, Rousou et al. undertook a matched prospective cohort study of 10 patients undergoing CABG [89]. Each patient had two portions of LSV harvested: one via EH and one via CT, and changes in functional proteins were measured using three techniques. Multi-photon imaging identified no observable breaks within the endothelium in either group. Immunofluorescence staining identified substantial membrane damage, reduced calcium mobilisation, lower caveolin, reduced VWF and lower NO production in response to bradykinin stimulation in the EH group. These results were corroborated via Western blot analysis and through observation of a higher mean esterase activity in the CT group indicating greater EC viability (*p* < 0.0001). Esterase activity reflects greater metabolic activity at the cellular level and may also represent improved cellular viability. Additionally, Lamm et al. [90] used scanning electron microscopy of LSVs harvested using EH or CT and found that EC integrity was dependent on the surgeons’ experience in using the EH technique, with improved EC integrity only identifiable when employed by a surgeon who had mastered the technique. Furthermore, they noted macroscopic differences between techniques with blood clots identifiable within EH veins more frequently than those harvested with CT. Furthermore, the VICO trial [91] analysed the impact of EH and CT on histological vein damage and its correlation with clinical outcomes. EC integrity, assessed by validated scoring of computerised assessment of CD34 staining, was better preserved in CT harvest groups compared to both closed and open tunnel EH (median % integrity 92.71% vs. 87.5% and 85.25%, *p* < 0.001). No differences in endothelial stretching were observed between groups. Greater hypertrophy of both circular and longitudinal vein wall muscle was observed following EH (*p* < 0.001) but this failed to demonstrate any direct correlation with clinical outcomes.

### 5.2. Storage Conditions

Once harvested, the LSV is stored in a preservation solution of the surgeon’s choice whilst the heart is prepared for grafting. The solution of choice is a spectrum of crystalloid to autologous blood solutions with various additives intended to buffer pH, confer osmotic impact, act as antioxidants, and mimic the normal composition of bodily intracellular fluid. During this time spent following harvest and prior to implantation, the LSV is in a period of relative ischaemia. The role of preservation fluids is therefore to attenuate EC damage where possible [92]. The primary source of ATP within the LSV is SMCs within the vein media and so the metabolic function of SMCs can be considered by the available concentration of ATP. Heparinised autologous whole blood (AWB) has been shown to better preserve ATP levels compared to normal saline. In fact, AWB has been shown in several studies to improve the vascular contractile reserve and EC function compared to normal saline [92,93,94] through maintenance of SMC tone and integral EC functions such as vasoactivity and platelet activation. Whilst heparin itself exhibits toxicity to the endothelium [95], the addition of whole blood appears to be protective which may be a consequence of its natural contents of energy sources, pH buffers, free-radical scavengers [93].

Papaverine is a commonly evaluated component of preservation fluid to induce SMC relaxation [96]. It is suggested that it may cause chemical damage to the endothelium due to its acidic pH and may also reduce local prostacyclin production. However, it has been demonstrated to reduce venospasm and EC preservation compared to AWB or isotonic crystalloids alone. The vein wall reacts to most local stimuli by undergoing prolonged muscular contraction, which disrupts the endothelium by reducing luminal surface area and causing herniation of SMCs into the lumen which, in turn, denudates the EC layer [97]. Increased SMC relaxation by vasodilators such as papaverine therefore mitigates venospasm, allowing the use of lower distention pressures during harvesting, thereby attenuating loss of ECs and EC damage.

More recently, new solutions have come to market. DuraGraft^®^ (Somahlution, Jupiter, FL, USA) is marketed as a one-time intraoperative treatment for the purposes of reducing vascular EC damage. It is a physiological salt solution (PSS) with additives to attenuate ischaemia and reperfusion injury [98]. Although, theoretically, solutions like these can make a difference, there is very limited data available to compare the efficacy of such solutions. A prospective ex vivo analysis [99] of 26 LSV segments (2 per patient) of which half (n = 13) had been preserved with DuraGraft^®^ and the other half with heparinised Ringer’s lactate for ≥90 min. Veins treated with DuraGraft^®^ demonstrated greater CD31 staining (*p* < 0.05) and reduced intimal swelling. There was also greater cell viability as demonstrated by Ki67 staining (*p* < 0.01) and analysis of cell death and senescence marker γH2AX demonstrated reduced numbers of arrested cells or cells with DNA damage in the EC layer of the DuraGraft^®^ group (*p* < 0.05). Furthermore, using phosphorylated-p53 as a marker of apoptosis, there was a significant reduction of cell death in veins treated with DuraGraft^®^ compared to Ringer’s lactate. Features of hypoxic stress were identified more frequently in non-DuraGraft^®^ treated conduits, although there was no clear reduction of ROS in this group. This study suggests that DuraGraft^®^ attenuates hypoxic injury and better preserves endothelium compared to Ringer’s lactate. Additionally, a comparison of PPS, AWB and DuraGraft^®^ to evaluate endothelial integrity of venous segments was undertaken by Toto et al. [100], which identified a statistically significant reduction in apoptotic cells in the DuraGraft^®^ storage sample after 2 h incubation versus PPS, with apoptotic cells identified using DNA fragmentation detection with fluorescein-12-dUTP. A further evaluation of DuraGraft^®^ by Tekin et al. [71] showed significantly lower total oxidative status (i.e., total oxidant molecules present) for DuraGraft^®^ stored samples, compared to both saline and AWB (*p* < 0.0001). The total antioxidant status (i.e., the ability to neutralise ROS) was lowest in the saline group and equal between both AWB and DuraGraft^®^. Combined, these findings suggest that the veins stored in DuraGraft^®^ had higher capacity to combat oxidative stress from ischaemia and therefore better protect EC function.

### 5.3. Pharma-Modulation

Pharmacological intervention plays a key role in secondary prevention of vein graft failure. Antiplatelet therapy has played a crucial role in improving outcomes after CABG and reducing vein graft failure, with administration of antiplatelets such as aspirin frequently utilised in clinical practice to mitigate complications following surgery [101].

#### 5.3.1. Aspirin

The long-term benefit of aspirin after CABG is mediated through reduction and prevention of platelet aggregation and a resultant reduction of thrombus formation, largely mediated by a reduction of thromboxane A2 (TXA2). Beyond platelet inhibition, aspirin also has a significant role in upregulation of EC NO synthesis [102]. The beneficial cardiovascular effects of aspirin are mediated through its inhibition of cyclooxygenase–1 (COX-1) activity. COX-1 is expressed within ECs and is responsible for the production of prostaglandins and resultant production of prostanoids such as TXA2. TXA2 stimulates platelet aggregation and the release of vasoactive pro-coagulant and pro-inflammatory factors; thus, many of the favourable effects of aspirin are attributed to the inhibition of TXA2. TXA2 has been shown to interfere with key endothelium-dependent pathways responsible for regulating blood flow via modulation of calcium activated potassium channels and impairment of cell signalling between EC and SMC layers, thereby impairing EC dependent vasodilation [103]. Aspirin has also been shown to evoke activation of eNOS and therefore increase NO synthesis. This occurs not only within ECs but within platelets themselves and appears to be independent of both COX-1 inhibition and TXA2 production. Two randomised studies have shown a significant increase in markers of NO formation (i.e., HO-1 and ADMA) in a non-dose-dependent fashion in patients with cardiovascular disease, supporting the hypothesis that an aspirin-mediated benefit occurs through the formation of NO, not just the commonly reported mechanism of platelet inhibition [104,105].

#### 5.3.2. Statins

Statins are predominantly utilised for their lipid-lowering properties. However, when given perioperatively to patients undergoing CABG, they have been shown to improve EC function, maintain NO levels and promote antioxidant activity, as well as inhibiting vasoconstriction, thrombosis, and the inflammatory response. Yang et al. [106] were able to demonstrate direct enhancement of saphenous vein EC expression of eNOS and subsequent increased NO production from ECs in response to statin therapy. A reduction in low-density lipoprotein (LDL) cholesterol has also been shown to result in reversal of EC impairment [107]. The clinical associations have also been proven—one of the first studies to randomise patients to statins after CABG was able to demonstrate that aggressive statin treatment (40 mg/day) after CABG was associated with less angiographic evidence of vein graft occlusion. Specifically, they identified an average of 10% increased vessel occlusion in patients undergoing aggressive statin therapy versus 21% increased occlusion in patients undergoing moderate (2.5 mg/day) statin therapy (*p* < 0.0001), as well as a reduced mean number of grafts exhibiting progression of atherosclerosis (25% in the aggressive treated group versus 39% in the moderate treated group, *p* < 0.001) [108].

#### 5.3.3. ACE Inhibitors/AngII Receptor Antagonists

Angiotensin-converting enzyme (ACE) is a key regulator of the renin angiotensin aldosterone system. Renin, which is synthesised as an inactive pre-pro-hormone, undergoes a proteolytic cascade resulting in its release into the system circulation as its active form. Within the circulation it acts upon angiotensinogen to generate angiotensin-I (AngI). AngI is subsequently cleaved by ACE to produce AngII. AngII is the primary atherogenic effector of the renin angiotensin aldosterone system, and activation of this system is associated with increased atherothrombotic events [109].

Many in vitro and in vivo animal studies have implicated AngII within pathways known to contribute to vein graft disease. AngII has been found to promote IH [110] and both SMC hypertrophy [111] and proliferation [112,113], and is evidenced to activate key inflammatory pathways which precipitate vein graft disease. Specifically, AngII has been demonstrated to stimulate release of IL6 from SMCs and macrophages [114] and to activate nicotinamide dinucleotide phosphatase oxidase with resultant production of ROS [115,116], thereby promoting oxidative stress within the vascular microenvironment. Angiotensin-converting-enzyme inhibitor (ACEi), Ramipril, assists in the preservation of EC function by inhibiting AngII production and attenuating the aforementioned pathways promoting inflammation and oxidative stress [117]. They also enhance NO production through prolongation of the half-life of bradykinin and stabilisation of the bradykinin receptor linked to the formation of NO [118]. Similarly, AngII type-1 receptor antagonists (ARBs) (i.e., losartan) exert similar benefit through competitive inhibition of AngII via its receptor.

Several human studies have evidenced improved arterial EC function following the administration of ACEi and ARBs in patients with diabetes [119] and hypercholesterolaemia [120]. Several large randomised studies including QUO VADIS [121] and IMAGINE [122] have looked at their clinical benefit after CABG but did not assess features of EC damage or post-operative vein graft patency directly. A 2005 blinded, randomised trial by Trevelyan et al. [123] demonstrated improvement in systemic endothelial function following pre-operative patient treatment with both ACEi (enalapril) or ARB (losartan). EC function, quantified by endothelial dependent flow mediated dilatation (FMD) of the brachial artery, increased from baseline in all groups (5.2% in surgery and enalapril group, *p* = 0.015 versus 5.0% in surgery and losartan group *p* = 0.0004 versus 3.0% in surgery alone group *p* = 0.05) at three months after CABG and was sustained at five months post-operatively. Interestingly all patients, even those who did not receive any pharma-modulation, demonstrated improved systemic endothelial function after CABG suggesting that coronary revascularisation alone confers some improvement of endothelial function.

Furthermore, research into lectin-like oxidised-LDL receptor-1 (LOX-1) has been shown to increase in response to ox-LDL and specifically in atherosclerotic plaques (such as those found in vein graft disease) [124,125,126]. These receptors have also been shown to upregulate in response to AngII through activation of AngII type-1 receptors, and interestingly the use of the ARB losartan inhibits LOX-1 upregulation [127,128,129]. Work by Ge et al. [130] utilising rabbit models identified that losartan treatment attenuated lesions and improved plaque stability; however, no reduction in intimal area was observed compared to the untreated group. LOX-1 expression was shown to increase in both the endothelium and lesion area of control mice, which appeared to be attenuated upon losartan treatment, suggesting that LOX-1 downregulation may confer benefits in the attenuation of vein graft disease.

### 5.4. Novel Approaches to Endothelial Preservation

Beyond the aforementioned approaches to endothelial preservation, several novel approaches are being considered including, but not limited to, gene therapy and utilisation of endothelial progenitor/colony-forming cells (EPCs/ECFCs).

Many of the most recent advances in gene therapy relating to vein graft disease have been reviewed in detail by Southerland et al. [131]. In brief, vein graft disease presents as a strong candidate for this treatment approach, predominantly due to the ability to administer the therapy ex vivo prior to reimplantation into the arterial circulation. This approach has the effect of ensuring localised delivery of the therapeutic whilst mitigating the effects of off-target or systemic effects.

There have been a significant number of pre-clinical studies undertaken to investigate the feasibility of gene therapy, with the primary targets relating to endothelial and smooth muscle cell preservation, and mitigation of thrombosis and inflammation. Approaches have included intraluminal or pressure-mediated transfection to induce eNOS overexpression [131,132], COX-1 overexpression [133,134], thrombomodulin and tPA overexpression [135,136], MCP1 inhibition [137,138], NF-κB inhibition [49,51], and E2F inhibition [139,140] among others. Of note, the E2F inhibition approach via oligodeoxynucleotide delivery had been expanded to human subjects (PREVENT trial) following success in pre-clinical models; however, outcomes of phase 3 have not shown significant differences in graft stenosis or patency by 12 months.

As an alternative approach, researchers are also investigating the potential of endothelial progenitor/colony-forming cells as a means of facilitating re-re-endothelialisation of saphenous vein grafts following implantation, with much of this branch of research having been concisely summarised by Paschalaki et al. [141]. In brief, EPCs categorise numerous populations of cells which express markers endothelial-specific markers which are known to contribute to vascularisation. ECFCs, as a subset, have been identified as having characteristics specific to endothelial origins, and these cells have been investigated in relation to a substantial number of vascular diseases, and have been considered in clinical applications such as gene therapy, vessel/tissue bioengineering, endothelial preservation, and repair.

With specific consideration of vein graft disease, circulating EPCs have been shown to regenerate graft endothelium in mice models, with approximately one-third of regenerated ECs derived from this circulating, bone-marrow derived population [142]. Furthermore, phenotypic analysis of peripheral blood derived ECFCs, compared to both human-derived arterial and venous ECs under high pulsatile flow conditions, depicted a highly proliferative, adaptable cell population which conforms to hemodynamic conditions, adopting a phenotype resembling that of the arterial ECs [143]. Interestingly, a study conducted by Feng et al. [144] was able to identify a therapeutic approach to increasing incorporation of ECFCs into vein graft endothelium through the topical treatment of high-density lipoprotein to the vessel adventitia. This treatment approach in mice showed a significant reduction in neointimal area (*p* < 0.001), improved blood flow (*p* < 0.0001), reduced inflammatory response (*p* < 0.05) determined by leukocyte adhesion, and enhanced endothelial regeneration (*p* < 0.05). This regeneration was attributed to improved ECFC migration and adhesion, which appear dependent on scavenger receptor class B type 1 expression, extracellular signal-regulated kinases and NO signalling.

Another study of note utilised human umbilical cord blood endothelial progenitor cells as a means of facilitating re-endothelialisation of vein grafts [145]. This group were able to identify that these cells exhibit superior adhesion capacity to cultured SMCs under shear stress conditions (0.5 dyne/cm^2^) compared to arterial ECs and peripheral blood ECs, believed to be the result of increased expression of cell surface α_5_β_1_ integrin. The proliferative capacity of these cells, to facilitate potential re-endothelialisation was also shown to be significantly faster under both shear conditions (0–15 dynes/cm^2^) and compared to arterial and peripheral blood ECs. Finally, using an immunodeficient vein graft mouse model, the group identified, with results indicating that these progenitor cells were capable of accelerating graft re-endothelialisation and, consequently, mitigating graft thrombosis. Whilst these results are promising, the haemodynamic conditions assessed here are significantly lower than those of the in situ arterial environment, and the highly thrombotic mouse model likely overestimates the anti-thrombotic nature of these cells. However, despite these drawbacks, EPCs present as a novel therapeutic approach to the preservation of endothelial function.

## 6. Conclusions

Despite extensive evidence of the superiority of arterial grafts, the LSV remains the most used conduits in cardiac surgery. Thus, it is essential to be able to prevent the major disadvantage of limited long-term patency (Figure 1). Understanding the mechanisms involved in the process of vein graft inflammation/IH is the first step toward the modification of the disease (Figure 2). Most of the current evidence is based on animal model studies that looked at an established disease using outdated assays and by doing this may have neglected for many years the role of ECs [146]. The simplistic idea that ECs are denuded and lost early, suggesting a limited role in the process of IH, needs to be challenged using more state-of-the-art assays such as single cell sequencing and looking at the acute activation of EC or phenotypic changes in response to exposure to acute arterial conditions on the LSV is grafted [147].

## Figures and Tables

**Figure 1 cells-11-03066-f001:**
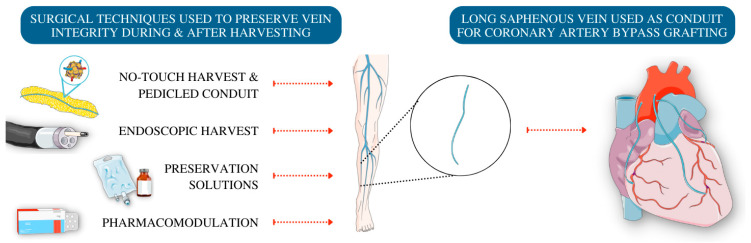
Overview of surgical techniques to preserve vein integrity pre and post harvesting.

**Figure 2 cells-11-03066-f002:**
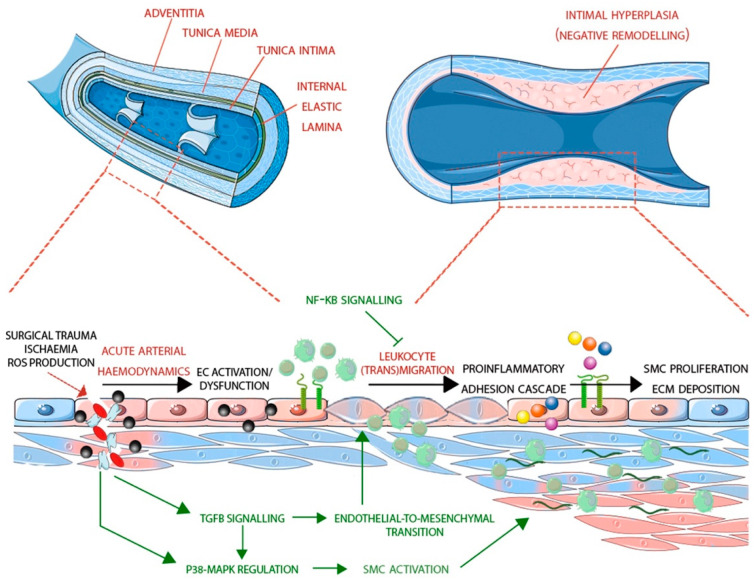
Overview of the onset and progression of vein graft disease, highlighting the signalling pathways implicated in the process at their location/time of induction.

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
