# Peer review of "The Role of Endothelial Cells in the Onset, Development and Modulation of Vein Graft Disease"

_cells, 2022, doi:10.3390/cells11193066_

Round 1

Reviewer 1 Report

This is an interesting review with a clear focus in identifying and describing the clinical challenge of the onset and progression of vein graft disease, in light of technical aspects regarding endothelial (dys)function.
The article is well written and addresses the technical aspects and concerns clearly, noting procedural, cellular and molecular aspects of the adaptation of venous EC to an arterial branch. 

I have very few comments on the content of this review, detailed below:

1. One suggestion is to drive the message home more clearly at the start of the paper, where the section on differences between venous and arterial EC could conclude with a clear spelling out of why this kind graft would naturally present with a physiological challenge. This kind of biological engineering involves morphological and metabolic shifts in highly specialised cells that require time, which results all too often in complications. One aspect that can be raised is the fact that endothelial cells from larger vessels are much less plastic than capillary endothelial cells, likely adding to the effectiveness of venous-to-arterial grafting. Maybe Figure 1 could be moved up to this section (2) to provide a visual aid to what the authors are dissecting later on.

2. Section 5.2 refers to storage solutions. Did the author's consider instead "storage conditions"? The metabolism, activation and function (barrier, endocrine, etc) of EC are heavily dependent on environmental cues, and while the composition of the storage solutions is mentioned in terms of pH and osmotic impact, one key aspect that is not mentioned is the oxygenation of these solutions. Venous EC encounter much lower O2 level in circulation than arterial ones, and their metabolism will be impacted by oscillations in oxygen availability, especially hyperoxidation. Are these solutions typically equilibrated to physiological O2 tensions during storage of venous grafts? And are there studies to investigate how venous EC adapt to typical arterial blood O2 and how long it would take? Would this provide an advantage to the engraftment process (i.e. allow adaptation to the adequate, physiological, arterial pO2?).

Finally, I find this review is ideally placed to expand on specific aspects of needed research that could lead to improvement of LSV grafts in the clinic. Pharma modulation is mentioned, but what about newer techniques in regenerative vascular physiology, such as using progenitor endothelial cells/ECFCs?

Overall very interesting article, and very simply a couple of omissions (hyperoxia and alternative correcting/regenerating strategies) could be included to make it more impactful, and likely relevant for a broader audience of researchers.

Author Response

Point 1: One suggestion is to drive the message home more clearly at the start of the paper, where the section on differences between venous and arterial EC could conclude with a clear spelling out of why this kind graft would naturally present with a physiological challenge. This kind of biological engineering involves morphological and metabolic shifts in highly specialised cells that require time, which results all too often in complications. One aspect that can be raised is the fact that endothelial cells from larger vessels are much less plastic than capillary endothelial cells, likely adding to the effectiveness of venous-to-arterial grafting.

Response 1: Thank you for this suggestion. We agree that clarification of the physiological challenges imposed by implanting venous ECs into arterial circulation, and the consecutive morphological and metabolic shifts required to ‘arterialise’ this graft are central to understanding how and why these grafts are wrought with complications. As such, we have expanded upon section 2 (differences between arteries and veins) to address this. This addition can be found on page 2 – 3, lines 82 – 99.

Point 2: Maybe Figure 1 could be moved up to this section (2) to provide a visual aid to what the authors are dissecting later on.

Response 2: We agree with this point and believe that moving the figures (particularly Figure 1) would improve the readability of our manuscript. As such, we have made amendments to the structure by moving the location of Figure 1 to the end of section 2 (page 3) to act as a primer for the following sections, and we have moved Figure 2 to the end of section 4 (page 6) to position it immediately following our text regarding the role of ECs in vein graft disease.

Point 3: Section 5.2 refers to storage solutions. Did the author's consider instead "storage conditions"? The metabolism, activation and function (barrier, endocrine, etc) of EC are heavily dependent on environmental cues, and while the composition of the storage solutions is mentioned in terms of pH and osmotic impact, one key aspect that is not mentioned is the oxygenation of these solutions. Venous EC encounter much lower O2 level in circulation than arterial ones, and their metabolism will be impacted by oscillations in oxygen availability, especially hyperoxidation. Are these solutions typically equilibrated to physiological O2 tensions during storage of venous grafts? And are there studies to investigate how venous EC adapt to typical arterial blood O2 and how long it would take? Would this provide an advantage to the engraftment process (i.e. allow adaptation to the adequate, physiological, arterial pO2?).

Response 3: Thank you for raising this point. We agree that “storage conditions” is better suited as a section heading, and have modified this accordingly. As you have stated environmental cues play a central role into the function, maintenance, morphology, and phenotype of endothelial cells. From a clinical perspective the preservation solutions used in maintaining vein graft patency prior to implantation are not inherently oxygenated to a physiological level, but rather are formulated in such a way as to provide an antioxidant effect to mitigate the effects of hypoxia between the point of extraction and implantation. Likewise, the efficiency of the surgical process is designed in such a way as to ensure that the graft is reintroduced to the circulation as expediently as possible to mitigate effects of hypoxia. As you have stated previously, the time required to facilitate a biomechanical shift in EC phenotype from a venous to arterial phenotype through environmental cues alone is extensive, and as such little research has investigated oxygenation of the preservation conditions to improving graft quality prior to implantation given the short turnaround in the surgical process.

There has been limited research into vein graft integrity in mice models (Parma et al. Prolonged Hyperoxygenation Treatment Improves Vein Graft Patency and Decreases Macrophage Content in Atherosclerotic Lesions in ApoE3*Leiden Mice. Cells 2020, 9, 336. https://doi.org/10.3390/cells9020336), which has shown some benefit in reducing ROS-induced macrophage accumulation in the plaque, but long-term hypoxia effects and angiogenesis remained. Attempts to improve oxygen availability in the human vein grafts have taken the form of surgical modifications (such as the no-touch technique and endoscopic harvesting, described in section 5.1., pages 6-8. Beyond this, only one article has investigated continuous perfusion of the saphenous vein with arterial oxygenated blood (Roshanali et al. Continuous Perfusion of Saphenous Vein by Oxygenated Blood during Beating Coronary Surgery. Int Cardiovasc Res J. 2015;9(3):177-179). Aside from stating good outcomes for patients, the authors provide no scientific or medically relevant analysis or results to draw conclusions from, and as such we have not included these aspects in this review.

Point 4: Finally, I find this review is ideally placed to expand on specific aspects of needed research that could lead to improvement of LSV grafts in the clinic. Pharma modulation is mentioned, but what about newer techniques in regenerative vascular physiology, such as using progenitor endothelial cells/ECFCs?

Response 4: Thank you for your support of our article. This work was originally intended to provide an overview of the clinical, or more specifically, practical implementation of these techniques in surgical practice. As such, we had originally chosen to omit some of the newer techniques that you have raised which have not been subject to clinical trials. Upon reflection though, we have decided that inclusion of these newer regenerative vascular physiology approaches would indeed provide a greater perspective on the clinical picture and what we might expect to see in practice in the future. To that end, we have included a subsection specifically addressing research into the use of aspects such as gene therapy and endothelial progenitor/ colony-forming cells (Section 5.4, page 11-12, lines 509 – 568). We have additionally included a reference provided by reviewer 3 relating to human umbilical cord blood–derived endothelial cells which we believe ties in well with this section (page 12, lines 554 – 568).

Reviewer 2 Report

Shameem S. Ladak in this review aims to provide an overview of the current working knowledge regarding the role of endothelial cells in the onset, development, and modulation of vein graft disease, as well as addressing current surgical and medical management approaches which aim to beneficially modulate endothelial function and improve patient outcomes.

Major points

No molecular mechanism / epigenetic modification was investigated to provide the “The Role of Endothelial Cells in the Onset, Development and 2 Modulation of Vein Graft Disease”.

A review should include an update on the state of the art. Too many old references are included in your manuscript.

The Figure are described only in the Conclusion section. However, the figures should guide the reader in understanding the manuscript.

The topics, certainly of scientific interest, were tackled without a critical spirit.

Author Response

Point 1: No molecular mechanism / epigenetic modification was investigated to provide the “The Role of Endothelial Cells in the Onset, Development and 2 Modulation of Vein Graft Disease”.

Response 1: We agree that expanding on the molecular mechanisms of vein graft disease in our manuscript would provide a greater overview of the disease pathology. As such, we have expanded upon section 4 to include these aspects. This includes an expansion of the mechanisms regarding NF-kB (page 4, lines 182-189), p38-MAPK (page 5, lines 201 – 219), and the inclusion of a reference from reviewer 3 regarding the role of PI3K/Akt signalling (page 5 – 6, lines 236 – 254).

However, we believe that we have already addressed specific epigenetic modifications relating to vein graft disease via our discussion of the arterial haemodynamic conditions (pressure, shear stress etc.) and how these changes can alter cellular phenotypes. Investigation of the specific mechanisms of this process (such as methylation and histone modification) have not been discussed in this review as there is limited to no research of these processes in vein graft disease pathology.

Point 2: A review should include an update on the state of the art. Too many old references are included in your manuscript.

Response 2: We agree that many older references have been utilised in this review. We have reviewed these older references and, in many cases, have updated or appended these with more modern equivalents. There are still older references that remain, in particular those related to: surgical techniques; studies related to those techniques; and clinical trials concerning surgical or pharmacological modification of surgical procedure. These references are still considered up to date, despite their age, as they are still referred to clinically to inform the surgical approaches and treatments utilised in practice, and no more recent articles have addressed these specific aspects in any greater detail than the originals.

Point 3: The Figure are described only in the Conclusion section. However, the figures should guide the reader in understanding the manuscript.

Response 3: We agree that the figures should be moved to and referenced earlier in the manuscript to improve the experience for the reader. To address this concern, we have moved our Figures prior to, or immediately following, their relevant section. In this case, we have moved Figure 1 to the end of section 2 (page 3) to act as a primer for the remainder of the manuscript, and we moved Figure 2 to the page 6 to act as a summary figure for the vein graft pathology process described above in Section 4.

Point 4: The topics, certainly of scientific interest, were tackled without a critical spirit.

Response 4: We are unclear on how exactly to address this point. While this review is inherently not systematic in nature, and thus not as critical as a review of that form, we believe we have written a manuscript which has provided a critical overview of the cellular and molecular aspects concerning endothelial cells in vein graft disease, and how these surgeons/ clinicians attempt to modulate and track these aspects to ensure appropriate diagnosis, prognosis, and treatment (as was outlined in the synopsis for this special issue).

Reviewer 3 Report

The main topic of the Review starts only at paragraph #4; the previous sections should be reduced, and the following expanded.

The following pertinent reports should be included in the review:

J Clin Med. 2022 Feb 18;11(4):1093. doi: 10.3390/jcm11041093.

Atherosclerosis. 2012 Jan;220(1):86-92.

Arterioscler Thromb Vasc Biol. 2010 Nov;30(11):2150-5.

Atherosclerosis. 2011 Feb;214(2):271-8.

PLoS One. 2014 Jun 2;9(6):e98904

Atherosclerosis. 2012 Apr;221(2):341-9.

Cardiovasc Res. 2014 Mar 1;101(3):513-21. doi: 10.1093/cvr/cvt333

Atherosclerosis. 2004 Dec;177(2):263-8.

Circ Res. 2000 Mar 3;86(4):434-40. doi: 10.1161/01.res.86.4.434.

Author Response

Point: The following pertinent reports should be included in the review:

J Clin Med. 2022 Feb 18;11(4):1093. doi: 10.3390/jcm11041093.

Atherosclerosis. 2012 Jan;220(1):86-92.

Arterioscler Thromb Vasc Biol. 2010 Nov;30(11):2150-5.

Atherosclerosis. 2011 Feb;214(2):271-8.

PLoS One. 2014 Jun 2;9(6):e98904

Atherosclerosis. 2012 Apr;221(2):341-9.

Cardiovasc Res. 2014 Mar 1;101(3):513-21. doi: 10.1093/cvr/cvt333

Atherosclerosis. 2004 Dec;177(2):263-8.

Circ Res. 2000 Mar 3;86(4):434-40. doi: 10.1161/01.res.86.4.434.

Response: Thank you for bringing our attention to these articles. While we have addressed many of the aspects that these articles are based on, we had not been specific enough to include these directly. As such, we have included the majority of these articles in our work at the following locations:

  • J Clin Med. 2022 Feb 18;11(4):1093. doi: 10.3390/jcm11041093. – This reference has been utilised to expand on our section regarding the comparisons of storage solutions. This reference can be seen on page 9, lines 409 – 413.
  • 2011 Feb;214(2):271-8 and Arterioscler Thromb Vasc Biol. 2010 Nov;30(11):2150-5. – Both of these references address endothelial progenitor cells (EPCs) and/or endothelial colony forming cells (ECFCs), and as such have been combined into a new section (section 5.4 – novel approaches to endothelial preservation). This section was created to address comments from reviewer 1 regarding a more general inclusion of EPCs/ ECFCs, and as such these references, among others, can be found on page 11 – 12, lines 509 – 568.
  • PLoS One. 2014 Jun 2;9(6):e98904 – This reference has been appended to our content on page 4, line 168 to address the variation in techniques seen in CABG surgery and in animal model studies.
  • 2012 Apr;221(2):341-9. – This reference has been used to create a new paragraph specifically regarding the role of PI3K/Akt signalling (and PTEN) in endothelial (dys)function. This can be seen on page 5, lines 236 – 254.
  • 2004 Dec;177(2):263-8. – This reference has been utilised to expand significantly on our discussion regarding pharma-modulation, in particular section ACE Inhibitors/ AngII receptor antagonists. This addition can be found on page 11, lines 498 – 508.
  • Circ Res. 2000 Mar 3;86(4):434-40. doi: 10.1161/01.res.86.4.434. – This reference has been appended to our section regarding EC activation and the resultant expression of molecules such as ICAM1, VCAM1 etc. that are responsible for the development of IH and vascular remodelling. This can be seen on page 4, line 163.

Round 2

Reviewer 2 Report

The authors partially met the reviewer's requests. 

The manuscript can be accepted in its present form

Reviewer 3 Report

-